# Effects of the Ingestion of Ripe Mangoes on the Squamous Gastric Region in the Horse

**DOI:** 10.3390/ani12223084

**Published:** 2022-11-09

**Authors:** Carolina J. F. L. Silva, Keity L. G. Trindade, Raíssa K. S. Cruz, Helena E. C. C. C. Manso, Clarisse S. Coelho, José D. Ribeiro Filho, Carlos E. W. Nogueira, Francesca Aragona, Francesco Fazio, Helio Cordeiro Manso Filho

**Affiliations:** 1Núcleo de Pesquisa Equina, Universidade Federal Rural de Pernambuco (UFRPE), Recife 52171-900, PE, Brazil; 2Faculdade de Medicina Veterinária, Centro Universitário Cesmac, Maceió 57051-160, AL, Brazil; 3Faculdade de Medicina Veterinária, Universidade Lusófona de Humanidades e Tecnologias (ULHT), Campo Grande 376, 1749-024 Lisboa, Portugal; 4Mediterranean Institute for Agriculture, Environment and Development, Universidade de Évora, 7005-869 Évora, Portugal; 5Departamento de Medicina Veterinária, Universidade Federal de Viçosa, Viçosa 36570-900, MG, Brazil; 6Departamento de Medicina Veterinária, Universidade Federal de Pelotas, Pelotas 96010-610, RS, Brazil; 7Department of Veterinary Sciences, University of Messina, Polo Universitario Annunziata, 98168 Messina, Italy

**Keywords:** equine, gastric ulcers, soluble carbohydrates, tropical nutrition

## Abstract

**Simple Summary:**

Gastric lesions can occur because of several factors, including nutritional issues. Horses throughout Brazil naturally consume fruits that are available in pasture areas (e.g., mangoes), mainly in times of low forage availability. Little is known about the effects of eating these foods on animal health; thus, a video endoscopy was performed to evaluate the results of mango consumption. Lesions were observed in the forms of hyperemia and erosions with the involvement of the gastric mucosa. In these cases, imaging exams are essential for diagnosis; the ideal prevention and treatment involve adequate nutritional management.

**Abstract:**

Erosions and gastric ulcers may be present in horses at any age and under different conditions of rearing and handling. In tropical regions, horses can feed on fruits rich in soluble carbohydrates, such as mangoes, but little is known about how these foods interact with their digestive systems. To test the hypothesis that the ingestion of ripe mangoes with peels could cause disturbances in the digestive processes of horses, an experiment was developed to monitor animals that had free access to ripe mangoes in their pasture areas. Horses (purebred Arabians, *n* = 5; ~340 kg, ~13 years) were evaluated by video gastroscopy and blood analysis. A controlled postprandial glucose curve for mango intake was also performed. Gastroscopies were performed at intervals of 15 days, starting in December, just before the beginning of the harvest, until the beginning of February, and days after the end of the harvest. Blood collection was performed on the same day between November and February for blood analysis. The results were submitted to ANOVA and Tukey’s test, with a significance level of *p* < 0.05. Gastroscopies indicated that four out of five horses had erosions and ulcers in the squamous region between 15 and 30 days after the start of the season. Biochemical tests indicated a reduction in plasma proteins during the harvest period, and the postprandial glucose curve showed concentrations above 200 mg/dL between 30 and 180 min after ingestion of 5.37 kg mangoes. The animals were not treated and recovered after 15 days of harvest and without ripe mangoes on the ground. It is concluded that the indiscriminate ingestion of mangoes favors the appearance of lesions in the gastric squamous region, to varying degrees, and that animals recover naturally after an average of 15 days from the end of the season when the animals return to their regular feeding with hay and grass pasture.

## 1. Introduction

Different studies have shown that horses can present erosions and gastric ulcers at any age and under different conditions of rearing and handling [1,2,3,4]. Such ulcers can be in the squamous or glandular regions, separated or grouped, and these types should be considered distinct entities of the same disease [2,5]. In general, stomach injuries may be associated with signs, such as colic, reduced sports performance, anorexia, weight loss or low body scores, loose stools, and behavioral changes [3,6,7]. The frequency of erosions and ulcerations described in the literature can vary from 10 to 90% in animals evaluated through gastroscopy and are associated with different epidemiological aspects [2,3,8,9], which makes prevention and diagnosis difficult.

Domestication and intensive management practices, especially in sports animals, have been linked to the appearance of erosions and gastric ulcers [1,2,3,5,10]. Horses under intense competition regimes may have more gastric ulcers, especially in the squamous region; when risk factors, such as intense training, are suspended or reduced, the amount and/or degree of injury may be reduced [5,11]. In addition, these groups of horses are more likely to have stress as the main trigger of these pathologies; therefore, it is suggested to discuss the triggering stress factors and how they may induce gastric ulcers in an athletic horse (physical fatigue, anxiety due to continuous work), such as for a horse kept in a box (lack of movement, sociability, nutrition) [1,2,3,4,12,13,14]. However, these lesions are also observed in animals kept on pasture with reduced intake or irregularity in forage availability [4], although less frequently than in athletic animals [2,9], showing that nutrition is an important factor for to reduce the incidence or grade of the lesions in horses under different management systems.

It is believed that the type of management of athletic horses, with less availability of forage throughout the day, and offering concentrates in spaced meals, leads to a reduction in the production and ingestion of saliva, decreases gastric pH, and makes the gastric environment more acidic, which may be predisposing factors for lesions [5,10]. Supplementation with high amounts of soluble carbohydrates and/or starch, which promote modification in the digestive process by a greater degree of short-chain fatty acid production, can stimulate the reduction of the local pH and modify the normal pattern of stratification of the pH in the stomach, producing erosions and ulcers in the squamous region and the *margo plicatus* [1,4,9]. The non-glandular represents nearly 1/3 of this organ and has minimal protective capacity from stomach secretions, differently from the glandular area. This last area represents around 2/3, with a larger capacity of protection in the pyloric region and medium protection capacity at the glandular fundus and cardia regions [4,9,11]. 

Due to this increase in fermentation and reduction in pH, it has been proposed to use nonpharmacological products that maintain the pH in the stomach close to or above 4, avoiding lesions in the squamous region [15,16]. It is observed that animals that consume more forage throughout the day have fewer gastric lesions, not only due to greater saliva production but also due to the continuous presence of food (forage) in the gastric lumen [4]. Thus, ulcers in horses are believed to represent a multifactorial process involving diet, exercise, medication, and other factors that need to be better understood [5,10,11]. The combination of good management practices that favor balanced nutrition and animal welfare can be important in the prevention and treatment of this disease in different groups of horses.

Horse breeding in Brazil extends across different regions, including the Atlantic Forest zone of Pernambuco. In this area, equid breeding has been practiced since the early 17th century, and animals have routinely consumed different types of local tropical fruits introduced over the years, such as mangoes (*Mangifera indica*) and jackfruit (*Artocarpus heterophyllus*), whether offered or present in the grazing areas, where animals consume them naturally, especially in the driest period of the year, between September and March, when there is less availability of forage. However, there is no information on the effects of mango consumption, which is a fruit rich in soluble carbohydrates (15%), 13.7% of which are sugars, and energy (60 kcal in 100 g), on the health of horses, differently from ruminants [17,18,19].

So, to test the hypothesis that the ingestion of ripe mangoes can produce changes in the digestive process in the stomachs of horses due to the number of soluble carbohydrates, a study was developed that aimed to evaluate the effects of the ingestion of ripe mangoes by horses during the collection period by means of videogastroscopies and blood analysis evaluations of horses in maintenance. 

## 2. Materials and Methods

This project was approved by the Ethics, Bioethics, and Animal Welfare Committee at the Faculty of Veterinary Medicine—Cesmac University Center (CIAEP/CONCEA), registration number 01200702819/2016-97—10 October 2016.

Animals and breeding system: In the current experiment, five Arabian mares were used, with an average body weight of 340 ± 5 kg, 13 ± 4 years of age; they were housed in Recife-PE (8°3′1″ S; 34°53′49″) throughout the summer (where there is one hour more of daylight). All horses were clinically analyzed by veterinarians weekly. The animals were kept in a semi-extensive system in Recife-PE (Brazil) (8°3′1″ S; 34°53′49″) with native pasture and supplemented with Tifton hay (*Cynodon dactylon*, 10.0 kg/animal/day). The pasture area had native trees, such as oitis (*Licania tomentosa*), paus d’arco (*Handroanthus serratifolius*), and mango trees (*Mangifera indica*). The period considered as mango harvest started at the end of November and lasted until the beginning of February, with a greater number of mature mangoes available on the ground from early December to mid-January.

The animals had free access to mineralized salt and water and did not receive concentrates. They remained free in the pasture for most of the day and were only taken to the stalls to receive Tifton hay (*Cynodon dactylon*), and were brushed and cleaned (7:00 a.m and 4:00 p.m.). The entire system of rearing and maintaining the experimental animals was regularly evaluated and adjusted, when necessary, through animal welfare assessment via the five domains program [20].

Clinical follow-up and video gastroscopies of horses: All horses were transferred to individual pens 15 h before all laboratory tests, received hay, and had at least 12 h of fasting for feeding. Laboratory evaluations were performed at 6 a.m., monthly, between November and February; collections were made in previously cooled vacuum tubes containing EDTA anticoagulants and immediately subjected to centrifugation to obtain the plasma, which was stored frozen (−20 °C). Subsequently, the plasma was submitted for analysis, i.e., of the plasma proteins and fibrinogen in semiautomatic equipment (Doles d250, Doles Equipamentos Laboratoriais, São Paulo-SP) using commercial kits. In addition, horses were evaluated by daily clinical examinations, as described by Speirs [12], to detect probable clinical alterations.

At the end of December, a post-prandial glucose curve was performed to evaluate the glycemic index of the ripe mangoes (peel and pulp) harvested fresh and freshly dropped on the ground. This process aimed to simulate the ingestion of mangoes by free animals in the paddock and obtain the post-prandial curve. For the composition of the post-prandial curves, horses were in individual boxes where they received the same number of calories from the mangoes. Blood samples were collected from the animals in the following periods: pre-test/fasting (T0) and at 30 (T30), 60 (T60), 90 (T90), 120 (T120), 180 (T180), and 240 (T240) minutes after the supply of 5.37 kg of ripe mangoes with peels but without the endocarp, corresponding to 3.22 Mcal/animal. The measurement of 60 cal/100 g of fruit was used according to the literature [17] to calculate calories. The blood samples were centrifuged, and the plasma obtained was frozen and subsequently subjected to the determination of glucose concentration (spectrophotometry) in semiautomatic equipment (Doles d250) using commercial kits.

Video gastroscopies were performed following the method described in the literature [2,21,22] using a 3-m video gastroscope (Tele-View USB Gastroscope, Advanced Monitors Corporation, San Diego, CA, USA). The examinations were performed on six occasions with 15-day intervals, as follows: preharvest 1 (30 days before), preharvest 2 (15 days before), harvest 1 (15 days of harvest), harvest 2 (30 days of harvest), postharvest 1 (15 days after), and postharvest 2 (30 days after). These evaluations were performed starting in December, just before the beginning of the harvest, until the beginning of February, days after the end of the harvest. This period’s mango harvest was determined according to the amount of ripe and fresh mangoes on the ground. 

Before the video endoscopies, the animals were fasted for 12 h, and on the day of the gastric exams, they were submitted for intravenous sedation with Detomidine (10 mcg/kg^−1^, Dormium V^®^, Agener União Saúde Animal, São Paulo, Brasil). For the exams, the stomach was inflated with an air pump, with pressure at 21 PSI for 2 min so that an adequate general visualization of the organ was possible [3]; 40 min after the examinations, access to food was again granted to the animals.

Statistical analysis: Statistical analysis was carried out using SigmaPlot 13.0 software (Systat Software, Inc., San Jose, CA, USA) for all analyses. One-way ANOVA and Tukey’s test with *p* set at *p* < 0.05 was applied for the evaluation of the statistical difference between different months on all blood parameters.

## 3. Results

The first two sequences of gastroscopic examinations in the preharvest showed that the animals did not present macroscopic lesions (Figure 1 and Figure 2); this finding is attributed to the fact that there were still no mature mangoes in the pastures to which the animals could access. At the end of the first half of December 2021 (harvest 1), a period where there was already great availability and consumption of ripe mangoes, two animals presented lesions, one classified as grade IV and the other as grade II, the latter having signs of hyperkeratosis. 

In January (harvest 2), three animals had lesions, two with grade IV lesions and one with grade III lesions. In the last assessments in January (postharvest 1), with no mangoes available on the ground, only one animal had grade III injuries. In February, no animal presented lesions, both around the greater curvature, *margo plicatus*, and in the lesser curvature, and the animals previously affected recovered without the need for therapeutic intervention. 

All lesions observed throughout the experimental period were in the squamous region and/or close to the *margo plicatus* (Figure 1 and Figure 2). During the follow-up period, the animals showed no clinical signs of gastrointestinal or locomotor disease, and no esophageal lesions were observed.

Hematological analyses did not reveal significant changes over the 4 months in which they were measured (Table 1). There was a decrease in the value of total plasma proteins in the period between December and January, in which the lowest concentration was detected (*p* < 0.05). No changes were observed in the fibrinogen concentration (*p* > 0.05). 

Regarding the glucose concentration obtained for the formation of the postprandial curve, an increase in glucose concentration was observed after the ingestion of fresh ripe mangoes (*p* < 0.001), with concentrations at T0 of 103.21 ± 10.50 mg /dL, reaching 279.36 ± 41.20 mg/dL at T30, remaining above 200 mg/dL until T180 (230.74 ± 68.70 mg/dL) (Figure 3). 

At the end of December, the whole ripe mangoes weighed an average of 254.30 ± 126.2 g. When the weight was measured without the seeds, the mean value was 215.30 ± 110.8 g. The pit represented approximately 15.3% of the fruit, as described in the literature [17]. The mango harvest lasted approximately 40 days, starting in the 1st week of December 2021 and ending at the end of the 1st fortnight of January 2022. It was relatively short due to the great loss of flowers and fruits (green) by the heavy rains in the early summer, but with large numbers of ripe mangoes falling to the ground daily. 

## 4. Discussion

During the experimental period, it was possible to observe that four of the five animals evaluated developed some type of erosion and/or ulcer at the squamous region in the stomach, most likely due to the large intake of ripe mangoes. However, throughout the analyzed period, no colic or other diseases were observed in the experimental animals. Other research showed that feeding has a great impact on the appearance of ulcerous lesions in the squamous region of the stomach of horses, and animals that consume larger amounts of foods rich in soluble carbohydrates may present more lesions of this type [4,5,11]. There are several proposals for the presence of gastric ulcers in the squamous region; it has recently been proposed that this type of lesion is more associated with the reduction of protective factors stimulated by increased stress [1] and that fermentable foods, such as concentrates rich in starch or other foods, can also trigger these lesions [23,24]. There are no data on the effects of mango ingestion by horses, but research with ruminants has shown that when mangoes are used in the diet of this group of animals, there is a great increase in gut fermentation, such as the ingestion of foods, e.g., corn, or foods rich in it, due to the high concentration of soluble carbohydrates [18,19]. 

Mangoes are rich sources of soluble carbohydrates, including starch, and have a high glycemic index. Research suggests that diets of concentrates rich in soluble carbohydrates produce some degree of gastric relaxation and increase the passage time of these foods through the organ [23,24]. This process extends the interval between the animals’ meals and favors the reduction of gastric pH. It is possible that the consumption of ripe mangoes can produce an effect similar to that observed in other species [13,18,19,23]. This fact can be corroborated by the evaluation of the postprandial glucose concentration, which detected concentrations above 200 mg/mL in the experimental animals for a long period (Figure 3), indicating a long digestion period of this fruit. This favors greater fermentation of nutrients in the stomach and the production of short-chain volatile fatty acids, lactic acid, and other compounds by the microbiota of the gastrointestinal tract [13,23]. These processes lead to prolonged periods of gastric acidity, alteration of the pH gradient, reduction of local protective factors, and deregulation in the concentration of hormones associated with food intake [1,7,23,24], culminating in the worsening of gastric lesions by stimulating longer periods of satiety, combined with lower forage intake.

The ingestion of ripe mangoes (possibly in large quantities) by the horses was a primary factor in the appearance of the ulcers visualized, and when the consumption of the fruits ceased, spontaneous healing of the lesions occurred. Spontaneous recovery after removal of the main causes for the appearance of gastric ulcers (e.g., supplementation with concentrates rich in soluble carbohydrates and intense exercise) was documented in the literature [4,9,11]. However, it should be kept in mind that fruit intake was not experimentally controlled and that animals may have consumed different amounts of mangoes throughout the period. Despite this, regular consumption of fruits was observed in all animals during the harvest.

In other species, mango ingestion is associated with an increase in adiponectin and cholecystokinin (CCK), which promote satiety for a longer period [25]. Moreover, bovines that are fed with mangoes showed large increases in ruminal fermentation with impacts on gut health [18,19]. It should also be noted that in horses, there is great diversity in the gastric microbiota, but most of the bacteria adhered to the mucosa and in ulcerative lesions are lactobacilli, streptococci, and lactic acid-producing bacteria of different species [23,26], which favor local fermentation and production of lactic acid and other short-chain fatty acids, reducing the local pH. Such observations, when compared with the findings of the present study, including the spontaneous recovery of lesions after the end of the mango season, confirm the possibility that the lesions observed in the animals developed because of the ingestion of ripe mangoes.

The most severe lesions were observed in the region of the lesser curvature when compared to the greater curvature and were always close to the *margo plicatus*. It is believed that increased or excessive consumption of mangoes modifies the digestive process in the stomachs of horses with an impact factor in the region of lesser curvature, where the pH gradient is narrower and more sensitive to injury [9,13,26]. Thus, gastric acid secretions from the glandular region and the fermentation of soluble carbohydrates reach areas with less protection more easily, promoting excessive acidification in the squamous region of the lesser curvature. Stratification by pH areas and greater protection of the surface tissue of the stomach are important for the gastric health of horses, and situations where there is a change in the pH gradient and loss of local protection favor the appearance of lesions in the stomach of these animals [9,11,13,26], as observed in the present research.

Different experiments have shown that supplementation with concentrates rich in soluble carbohydrates produces ulcers in the squamous region of the stomach [1,5,11,21] and, in a situation similar to that described in the present work, is correlated with the presence of ulcers. The use of hay alfalfa or continuous forage feeding has been shown to combat stomach acidification by improving chewing and saliva production [13,14,15,23,25,26,27]. 

The constant presence of food in the stomach can help maintain the normal acid gradient [4,9]. The use of supplements that maintain or restore the pH in the region, and that are rich in pectin and lecithin incorporated in the concentrated ration [28], or even combinations of foods and nutrients in supplements that aim to reduce ulcers [29], have been suggested. However, in this study, the animals were not treated, and only the primary factor (mangoes) was removed together with the supply of hay and natural forage. This management was enough for the total remission of the visualized lesions to occur, with spontaneous healing between 15 and 30 days after the animals returned to their regular feeding patterns. Therefore, the maintenance of dietary programs with high-fiber diets that reduce stomach injuries in the long term should be used to combat the disease [4] since medicines cannot always be used for long periods without undesirable effects [30].

Clinical and complementary laboratory tests on experimental animals did not show alterations indicative of lesions in the stomach, which was expected since it is already shown that gastroscopy is the only exam that can provide the definitive diagnosis of the disease [2,4,9,23]. In biochemical tests, only the plasma protein concentrations showed a significant reduction in the period where there were ripe fruits on the ground, which may indicate a disruption in the digestive process and the occurrence of ulcers in the squamous region caused by the excessive ingestion of mangoes since this parameter is indicative of health in the digestive process.

Recently, rapid stool blood and albumin tests became available on the market; these molecules should not be present in the feces unless there is some type of injury along the digestive tract, such as gastric and/or colonic ulcers [31,32]. Thus, these rapid tests could serve as quick screening tests for video endoscopies to be performed later, as they are invasive tests and not without operational risks [33].

However, there are few reports of hypoproteinemia and/or anemia in animals affected by gastric ulcers [3,23], but sequential tests may be necessary for this type of diagnosis. In the current experiment, the animals were monitored monthly, and a reduction in the concentration of plasma protein was observed, but no changes were observed in the other blood parameters of the animals (Table 1). More research accompanied by a greater frequency in the determination of plasma proteins in the blood may be necessary to indicate the real importance of these findings and, thus, confirm the validity of this test as a parameter to be requested for monitoring animals with suspected intestinal lesions, such as gastric and colonic ulcers. 

## 5. Conclusions

Ingestion of ripe mangoes ad libitum in large quantities caused the appearance of erosions and/or gastric ulcers in horses, and after the exclusion of this food at the end of the season, spontaneous healing of the lesions occurred. These findings corroborate the idea that the incidence of gastric ulcers in athletic horses, when fed a large number of soluble carbohydrates, often present gastric ulcers, but the reduction of foods with this characteristic can have a positive impact on the horse’s health, especially when combined with a reduction in the typical stress surrounding training and competition venues. Our experimental horses were not subjected to any major stresses and their handling practices were in accordance with the five domains of the welfare program.

Finally, this research, despite demonstrating the effects of the improvements in long-term feeding management through the reduction of the supply of soluble carbohydrates, has some limitations, among them the reduced number of animals and the reduced control of food intake. Other inflammation and stress biomarkers could also be evaluated to better understand the process observed in the current experiment. Thus, we hope that other projects can be developed to better understand the long-term feeding effects of animals and stomach diseases.

## Figures and Tables

**Figure 1 animals-12-03084-f001:**
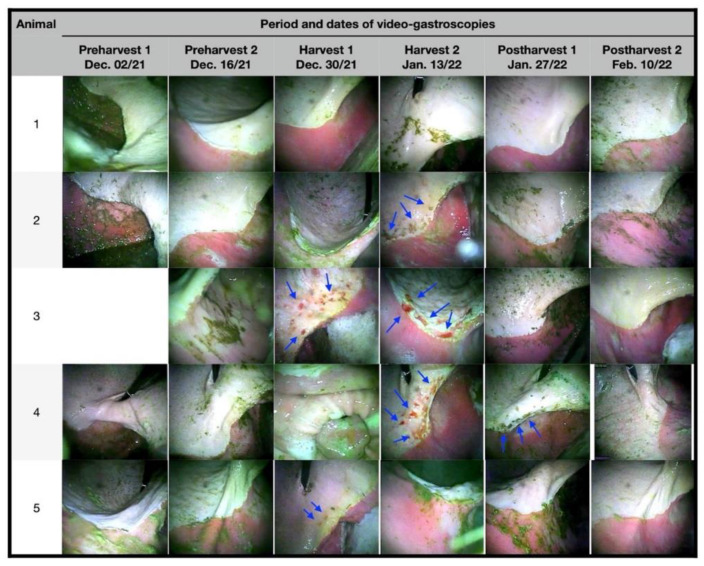
Photographs of video gastroscopies of experimental animals preharvest (1 and 2), harvest (1 and 2), and postharvest (1 and 2) in the region of the lesser curvature, close to the antrum of the pylorus. Note: Image of the 3rd animal was not taken at preharvest 1. Numbers at 1st column are experimental animals’ numbers. Arrows indicate ulcers and erosions area.

**Figure 2 animals-12-03084-f002:**
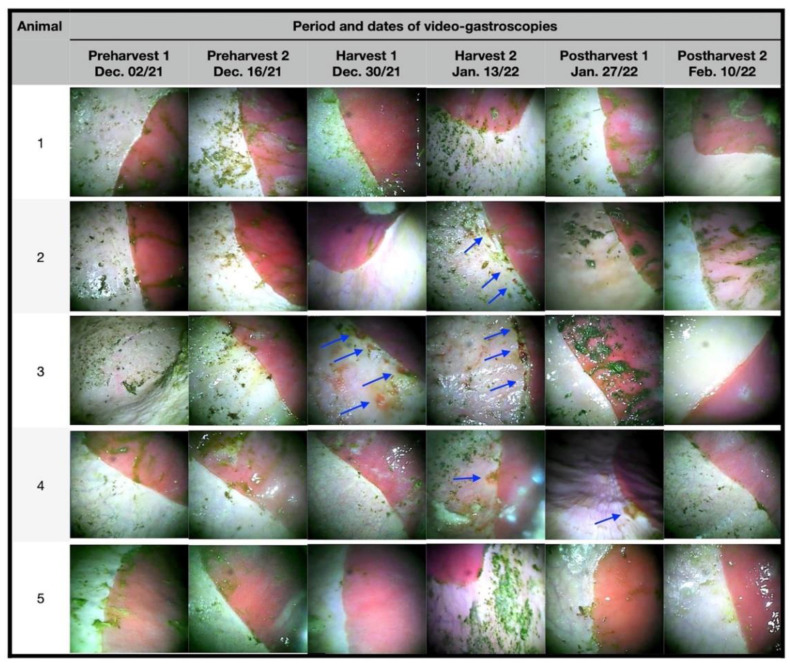
Photographs of the video gastroscopies of experimental animals preharvest (1 and 2), harvest (1 and 2), and postharvest (1 and 2) in the greater curvature region of the squamous and *margo plicatus*. Numbers at 1st column are the experimental animals’ numbers. Arrows indicate ulcers and erosions areas.

**Figure 3 animals-12-03084-f003:**
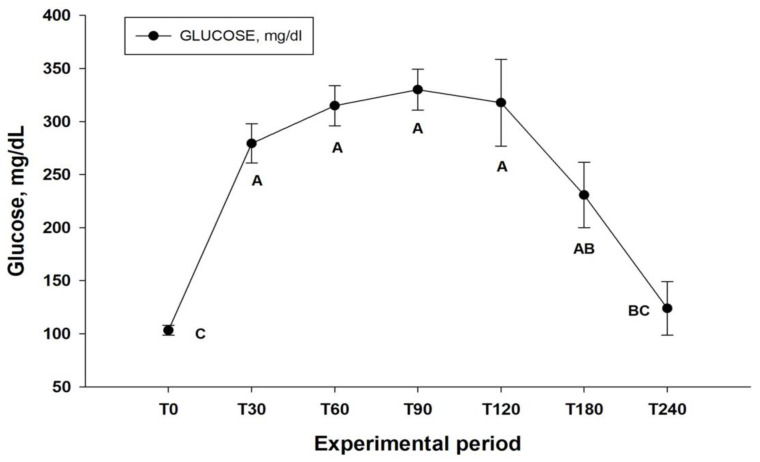
Plasma glucose concentration before and after the ingestion of ripe mangoes with peels for the characterization of the postprandial curve of the mangoes. Different letters indicate that *p* < 0.05 by Tukey’s test.

**Table 1 animals-12-03084-t001:** Results of blood and biochemical analyses in experimental horses between November and February.

Parameters	Month			
	November	December	January	February
Red cells, ×10^6^/uL	7.30 ± 0.38	6.86 ± 0.26	7.18 ± 0.09	6.49 ± 0.36
Hemoglobin, g/dL	10.80 ± 0.49	10.24 ± 0.31	10.62 ± 0.36	9.58 ± 0.29
Hematocrit, %	31.74 ± 1.58	29.94 ± 0.93	31.34 ± 1.05	27.94 ± 0.89
VCM, fL	43.54 ± 1.25	43.78 ± 1.44	43.66 ± 1.47	43.30 ± 1.30
CHCM, g/dL	34.08 ± 0.43	34.20 ± 0.12	33.88 ± 0.22	34.30 ± 0.21
RDW-SD, fL	34.28 ± 0.49	34.54 ± 0.44	35.10 ± 0.24	35.44 ± 0.24
RDW-CV, %	19.60 ± 0.47	19.90 ± 0,63	20.24 ± 0.81	20.80 ± 0.84
Platelets, ×10^3^/µL	157.60 ± 27.19	177.00 ± 13.58	177.60 ± 11.67	211.20 ± 30.17
White blood cell, ×10^3^/µL	7.92 ± 0.76	8.70 ± 1.35	6.90 ± 0.55	7.34 ± 0.86
Lymphocytes, ×10^3^/µL	3.40 ± 0.55	2.72 ± 0.26	2.56 ± 0.39	2.74 ± 0.54
Other white cells ×10^3^/µL	4.52 ± 0.44	5.98 ± 1.12	4.34 ± 0.51	4.60 ± 0.38
Plasma protein, mg/dL	7.00 ± 0.11 ab	6.64 ± 0.16 bc	6.32 ± 0.14 c	7.17 ± 0.08 a
Fibrinogen, mg/dL	0.36 ± 0.04	0.24 ± 0.04	0.20 ± 0.001	0.32 ± 0.05

Note: different letters on the same line indicate *p* < 0.05 by Tukey’s test. Other white cells: segmented, rods, eosinophils, basophils.

## Data Availability

Not applicable.

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
