# Peer review of "Effects of the Ingestion of Ripe Mangoes on the Squamous Gastric Region in the Horse"

_animals, 2022, doi:10.3390/ani12223084_

Round 1

Reviewer 1 Report

This paper is a useful investigation into the effect of mango ingestion on gastric ulceration in horses. The numbers of horses are very few and therefore caution should be used in application of these results. The authors need to acknowledge this in the manuscript and indeed present a power calculation to show the likely strength of the data collected. The discussion is over-long and requires editing as per the comments below.

Page

Comment

Abstract

5% ? need to state that significance level was set at P<0.05

Introduction

80-81

Mean decreases the gastric pH i.e. making condition more acidic

91

Spelling -  pyloric

Material and Methods

179

As before change 5% to P<0.05

Results

ok

Discussion

274

Bovines that are…and impact on

This section is over-long and delves into areas such as stool tests as indicators of gastrointestinal ulceration. While useful information it bears no relationship to the results and should be omitted. So omit 325-333. Section 305=315 is also not directly relevant to the results presented here and should be shortened or omitted.

Author Response

We thank the reviewer for helpful comments, we have improved the manuscript as suggested by including all the changes of the table shown by review in the revised manuscript in red color text.

Reviewer 2 Report

I appreciate all the work that went into this study. However, the authors can need to improve some aspects.

- English need to be revised.

- Tittle: it is a little confused. It seems that the article explain how to treat gastric ulcerations with ripe mangoes. The tittle can be modified as: “ Effects of the ingestion of ripe mangoes on the squamous gastric region in horses”

- Lines 86-87: These authors comments that horses’ stomach can be divided in two major regions. Nowadays it is well known that the equine stomach is divided in two major regions. Please, remove can be from the sentence.

- Lines 110-111: a fruit rich in soluble carbohydrates (15%; sugars 13.7%). What it is corresponding to 15%? Soluble carbohydrates? The 13.7% of sugars came from the 15% of soluble carbohydrates or are separated?

- Lines 113-114: To test the hypothesis that the ingestion of ripe mangoes can produce changes in the digestive process in horses’ stomach increasing local fermentation process. This sentence is grammatically incorrect. Please rewrite it.

- Materials and Methods. The authors described how they evaluate the glycemic index but it is not very clear if the animals were together in the same paddock or they were separated. How the authors knew each horse ate the same amount of mangoes if they were together?

- Lines 194-195: with no sleeves 194 available on the ground. Sleeves? Maybe the correct word is mangoes?

- It could be nice if the authors included the limitations of the study at the end of the discussion.

- Lines 353-355: “In addition, this research showed that video gastroscopy performed every 15 days was able to demonstrate the evolution of gastric lesions caused by a food with a higher content of soluble carbohydrates and its natural resolution after the removal of this food from the equine nutrition.” This have been previously describe so it is preferable to remove this sentence from the conclusions of the study.

Author Response

I appreciate all the work that went into this study. However, the authors can need to improve some aspects.

Author's reply: We thank the reviewer for the helpful comments, we have improved the manuscript as suggested in the revised manuscript in red color text . Below I reported all reply of each comment:

- English need to be revised.

Author's reply: the English language was improve as requested

- Tittle: it is a little confused. It seems that the article explain how to treat gastric ulcerations with ripe mangoes. The tittle can be modified as: “ Effects of the ingestion of ripe mangoes on the squamous gastric region in horses”

Author's reply: the Title changed as requested

- Lines 86-87: These authors comments that horses’ stomach can be divided in two major regions. Nowadays it is well known that the equine stomach is divided in two major regions. Please, remove can be from the sentence.

Author's reply: It is removed

- to 15 Lines 110-111: a fruit rich in soluble carbohydrates (15%; sugars 13.7%). What it is corresponding %? Soluble carbohydrates? The 13.7% of sugars came from the 15% of soluble carbohydrates or are separated?

Author's reply: Yes it is correct. Soluble carbohydrates is 15% and for 13.7% is sugars

- Lines 113-114: To test the hypothesis that the ingestion of ripe mangoes can produce changes in the digestive process in horses’ stomach increasing local fermentation process. This sentence is grammatically incorrect. Please rewrite it.

Author's reply: It is corrected as suggested and it is rewrite

- Materials and Methods. The authors described how they evaluate the glycemic index but it is not very clear if the animals were together in the same paddock or they were separated. How the authors knew each horse ate the same amount of mangoes if they were together?

Author's reply: The horses were separated and it was explained in the text

- Lines 194-195: with no sleeves 194 available on the ground. Sleeves? Maybe the correct word is mangoes?

Author's reply: It is corrected

- It could be nice if the authors included the limitations of the study at the end of the discussion.

Author's reply: Insert it

- Lines 353-355: “In addition, this research showed that video gastroscopy performed every 15 days was able to demonstrate the evolution of gastric lesions caused by a food with a higher content of soluble carbohydrates and its natural resolution after the removal of this food from the equine nutrition.” This have been previously describe so it is preferable to remove this sentence from the conclusions of the study.

Author's reply: It is removed